# *DLX* Genes in the Development and Maintenance of the Vertebrate Skeleton: Implications for Human Pathologies

**DOI:** 10.3390/cells11203277

**Published:** 2022-10-18

**Authors:** Giovanni Levi, Nicolas Narboux-Nême, Martine Cohen-Solal

**Affiliations:** 1Physiologie Moléculaire et Adaptation, CNRS UMR7221, Team BBC, Département AVIV, Muséum National d’Histoire Naturelle, 75005 Paris, France; 2Bioscar INSERM U1132, Université Paris Cité, Hôpital Lariboisière (APHP), 75010 Paris, France

**Keywords:** *Dlx* genes, skeleton, bone, osteoblasts, chondrocytes, osteoclasts, differentiation, homeobox genes

## Abstract

Skeletal shape and mechanical properties define, to a large extent, vertebrate morphology and physical capacities. During development, skeletal morphogenesis results from dynamic communications between chondrocytes, osteoblasts, osteoclasts, and other cellular components of the skeleton. Later in life, skeletal integrity depends on the regulatory cascades that assure the equilibrium between bone formation and resorption. Finally, during aging, skeletal catabolism prevails over anabolism resulting in progressive skeletal degradation. These cellular processes depend on the transcriptional cascades that control cell division and differentiation in each cell type. Most *Distal-less* (*Dlx*) homeobox transcription factors are directly involved in determining the proliferation and differentiation of chondrocytes and osteoblasts and, indirectly, of osteoclasts. While the involvement of *Dlx* genes in the regulation of skeletal formation has been well-analyzed thanks to several mutant mouse models, the role of these genes in the maintenance of bone integrity has been only partially studied. The importance of *Dlx* genes for adult bone tissues is evidenced by their central role in the regulatory pathways involving *Osx/Sp*7 and *Runx2*, the two major master genes of osteogenesis. *Dlx* genes appear to be involved in several bone pathologies including, for example, osteoporosis. Indeed, at least five large-scale GWAS studies which aimed to detect loci associated with human bone mineral density (BMD) have identified a known *DLX5/6* regulatory region within chromosome 7q21.3 in proximity of SEM1/FLJ42280/DSS1 coding sequences, suggesting that *DLX5/6* expression is critical in determining healthy BMD. This review aims to summarize the major findings concerning the involvement of *Dlx* genes in skeletal development and homeostasis and their involvement in skeletal aging and pathology.

## 1. Skeletal Development and Maintenance

The skeletal system determines vertebrate morphology and physical capacities; its multiple functions include support of the body weight, enabling locomotion, protection of internal organs (such as the brain and thoracic organs), and assuring mineral homeostasis. The shape of bones and cartilages is genetically determined during embryonic development and is then refined during postnatal growth, adapting to environmental constraints and mechanical forces. Later, during adult life, skeletal growth and maintenance depends on genetic, environmental, hormonal, dietary, and mechanical factors. The highest bone mass level (known as peak bone mass) is reached during puberty for humans at about 20y of age and is subsequently maintained thanks to cellular and molecular regulatory systems, which ensure an equilibrium between the processes of bone formation and bone resorption, resulting in bone homeostasis. This balance relies on continuous communication between osteoblasts and osteoclasts. Pathological bone conditions can result from an imbalance in osteoblast–osteoclast communication. For example, during aging, bone catabolism prevails over bone anabolism with a decrease in skeletal mass leading to osteoporosis and higher risk of fractures. A good understanding of how osteoblasts, osteoclasts, and their precursors mutually regulate their rates of proliferation and differentiation is, therefore, the starting point to tackle bone conditions such as osteopenia and osteoporosis that are becoming more prevalent as human life expectancy increases. 

Three different cell populations participate in the formation of mammalian bones [1]: (1) cranio-facial bones are formed by cranial neural crest cells (CNCCs), (2) the axial skeleton of the trunk and some components of the skull derive from the paraxial mesoderm, and (3) the appendicular limb skeleton derives from the lateral plate mesoderm. Independently from their embryonic origin, these mesenchymal cells differentiate into osteoblasts endowed with the capacity to secrete bone matrix. 

Two transcription factors are the major players of the osteoblast differentiation program: the Runt-related transcription factor 2 (Runx2) [2] and the zinc finger-containing Sp7/Osterix [3]. Indeed, targeted inactivation of either *Runx2* or *Sp7* in the mouse results in complete absence of bone formation due to the lack of osteoblast maturation [2,3,4,5]. Mutation of either of these two genes in humans results in abnormal bone formation [6,7]. During osteoblast differentiation, mesenchymal progenitors first get committed to the osteoblastic lineage through the activation of *Runx2*, then the coactivation of Sp7 defines the transition to osteoblast precursors and finally to mature osteoblasts. In *Sp7*-null mice, osteoblast precursors do not differentiate into mature osteoblasts even in the presence of Runx2 while, in the absence of *Runx2*, *Sp7* is never activated [3]. Therefore, *Sp7* is downstream of Runx2 in the osteoblast differentiation cascade.

There are two main modes through which bone is formed: (1) the process of “intramembranous ossification” which represents the direct differentiation of mesenchymal cells to osteoblasts, and (2) that of “endochondral ossification” which involves the progressive replacement of a cartilaginous template by osteoblasts and osteocytes [1]. While intramembranous ossification occurs during the formation of flat bones, such as those of the skull, endochondral ossification is the main developmental strategy of most components of the appendicular and axial skeleton of vertebrates. 

Intramembranous ossification involves the bone morphogenetic protein (BMP)-dependent activation of *Runx2* whose expression, in flat bones, is limited to mesenchymal cells of the osteoblastic lineage [4,8]. Runx2, in turn, activates *osx, osteocalcin*, *osteopontin*, and other bone-specific genes. Intramembranous ossification is the principal bone formation process for most CNCC-derived bones.

Endochondral ossification begins with the condensation of mesenchymal cells triggered by cell–cell and cell–matrix interactions in areas of the embryo which will give rise to long bones [9]. Sox9, the master regulator of chondrocytic differentiation, is the main transcription factor activated during endochondral mesenchymal cell condensation. Sox9 is a member of the HMG (high mobility group) protein family and binds directly to HMG sites present in regulatory regions of some of the main players of cartilage differentiation, including type II and type XI collagens [10]. Cells located in the center of the condensing regions differentiate into chondroblasts, while those at the periphery form the perichondrium. After proliferation, chondroblasts differentiate into prehypertrophic chondrocytes under the control of Wnt4, produced by cells bordering the periarticular perichondrium [11]. After the cartilage model is formed during fetal life, it produces the signaling protein Indian hedgehog (IHH) which, in turn, induces the secretion of parathyroid hormone related protein (PTHrP). PTHrP then binds to receptors on chondrocytes of the periarticular perichondrium [12] maintaining their proliferative status and preventing their differentiation into post-mitotic, hypertrophic chondrocytes. Therefore, the feedback loop between PTHrP and IHH regulates the rate of chondrocyte differentiation and determines the sites in the perichondrium where the first differentiation in osteoblasts occurs. 

Bone morphogenetic proteins (BMPs), members of the TGFß superfamily of secreted growth factors, are expressed in the entire developing skeleton and play critical roles in cell differentiation, cell type specification, and apoptosis during endochondral ossification.

BMPs induce apoptosis in cells surrounding the perichondrium and delineate the shape of the cartilage bone template [13]. Aside from its role in osteoblast differentiation, Runx2 also plays an important role in chondrocyte maturation to the hypertrophic phenotype [14].

Fully differentiated hypertrophic chondrocytes express type X collagen, alkaline phosphatase, and proangiogenic factors such as VEGF that stimulate the vascular invasion of the perichondrium and of the hypertrophic zone. The subsequent colonization of the cartilaginous template by osteoprogenitors and preosteoclastic cells arriving through the vascular system is then the critical step allowing endochondral ossification. These progenitors give rise to the two major cell types of the bone: the bone-forming osteoblasts and the bone-resorbing osteoclasts. To assure bone homeostasis, osteoblasts and osteoclasts need to maintain a tight mutual regulation through direct cell-to-cell contact or through signals mediated by secretory proteins. The osteoblast/osteoclast direct interaction allows bidirectional signaling through FASL-FAS, EFNB2-EPHB4, or SEMA3A-NRP1, leading to the mutual regulation of differentiation and survival. Alternatively, molecules secreted by osteoblasts, including for example RANKL/OPG, M-CSF, WNT5A, and WNT16, modulate osteoclast differentiation and development [15,16].

As mentioned above, Runx2 is required for early commitment of mesenchymal cells to the chondrocytic and osteoblastic lineage [2,4,5]. *Runx2*-null mice do not form normal hypertrophic chondrocytes or osteoblasts and their cartilaginous skeleton remains uncalcified. *Osx* and *Atf4* are downstream of Runx2 and participate to the osteoblastic phenotype. *Osx-*null mice can form normal hypertrophic chondrocytes but not normal osteoblasts [3], while *Atf4*-null mice display reduced osteoblastic activity and abnormal skeletal growth [17]. Many other transcription factors are also involved in the regulation of osteoblast differentiation, including, for example, Msx1 [18], Msx2 [19], Twist [20,21], AP1(Fos/Jun) [22], Krox20 [23], Sp3 [24] and, as we will see in this review, members of the Dlx homeobox family. Understanding how the regulatory networks involving these transcription factors affects bone formation and homeostasis is one of the major challenges in the field of bone biology.

## 2. The *Distal-Less* Gene Family in Vertebrates

The *Distal-less* (*Dll)* gene of *Drosophila* encodes a homeodomain protein that is one of the first to be expressed during leg primordia and cephalic development [25,26,27]. Adult *Dll Drosophila* mutants present reduction and dysmorphogenesis of distal segments of most appendages, including legs, antennae, and mouth parts, indicating that the *Dll* activity is required for appendage proximo-distal (PD) organization during early larval stages [25,28,29]. In vertebrates, *Dlx* genes share a homeodomain, homologous to that of *Drosophila Dll*.

Phylogenetic analyses suggest that, during evolution, a linked pair of *Dlx* genes was generated through ancestral tandem duplication after the divergence of chordates and arthropods, but prior to that of vertebrates. This bigenic *Dlx* cluster was then duplicated twice at the same time as the two genomic duplications that led to the four *Hox* clusters of existing vertebrates. Finally, the pair of *Dlx* genes linked to the *HoxC* cluster was lost in mammals [30,31]. Therefore, the present mammalian genome includes six *Dlx* genes linked in three bigenic pairs, in a tandem convergent configuration facing each other through the 3′ end: *Dlx*1/*Dlx*2; *Dlx*3/*Dlx*4 (the latter previously named *Dlx7*) and *Dlx*5/*Dlx*6 [27]. Interestingly, similarly to the fact that the Drosophila *Dll* gene is located in proximity of the HOM-C complex, in mammals, the three *Dlx* bigenic clusters are located in the same chromosomes where the HOX complexes are found. For example, in humans, *DLX3* and *DLX4* are located on chromosome 17q21 [32,33,34] together with the *HOX-B* gene cluster, *DLX1* and *DLX2* genes are associated with the *HOX-D* gene cluster on chromosome 2 [35], while the *DLX5* and *DLX6* genes are linked to the *HOX-A* cluster on chromosome 7 [36]. 

All vertebrate *Dlx* genes present a similar exon-intron structure with three or four coding exons [31,35,37,38]. The first exon of human and mouse *DLX6* contains a CAG/CCG (poly-glutamine/poly-proline) repeat region with a polymorphic length (CAG_12–20_) similar to the trinucleotide repeat region of the Huntington’s disease gene [37].

The expression pattern of each *Dlx* gene in a bigenic pair is similar, but not identical [31,36,39,40,41,42]: during early development, all *Dlx* genes are expressed in the pharyngeal arches, in the limbs and in the brain in overlapping territories of expression which define presumptive regions of the adult. For example, a complex partially-overlapping combinatorial code of *Dlx* gene expression in the pharyngeal arches determines the identity and the shape of each craniofacial bone [43]. These overlapping expression patterns might arise from the fact that linked *Dlx* genes share cis-acting regulatory sequences [40,44,45] and form the capacity of *Dlx* genes to mutually regulate or auto-regulate their expression [42,44,46].

Aside from classical transcriptional regulation, *Dlx* genes are subjected to several other levels of regulation. In particular, the *Dlx*5/6 locus is imprinted fully in humans [47], but only partially in the mouse [48,49]. DNA looping and DNA methylation are important in the control of *Dlx*5 and *Dlx*6 expression and it has been suggested that defects in these regulations might contribute to the origin of Rett syndrome [49]. Furthermore, a non-coding RNA, called *Dlx6-AS1* (*Evf-2*), is transcribed from the *Dlx*-5/6 ultra-conserved region and cooperates with the Dlx-2 protein to increase the transcriptional activity of *Dlx*-5/6 enhancers [50]. Alongside acting as transcription factors, Dlx proteins might also have other regulatory functions by participating in molecular complexes in the cytoplasm. For example, alternative splicing generates a Dlx5 variant that maintains the N-terminal region but lacks the homeodomain; this splice variant is mostly localized in the cytoplasm. The N-terminal region of Dlx5 interacts with the adaptor protein Melanoma Antigen Family D1 (MAGE-D1). Dlx5/MAGED1 interaction is important to modulate the role of Dlx5/6 in bone formation [51] through the formation of protein complexes that determine the cellular and/or transactivating activity of Dlx5. In particular, it has been shown that the neurotrophin receptor-interacting protein Necdin can promote osteogenic differentiation by forming a complex with MAGE-D1 and Dlx5, to inhibit proliferation and death of osteogenic cells [52].

These different levels of regulation might also act simultaneously in the adult and should be taken into account to understand the Dlx-dependent regulations in bone.

## 3. *Dlx* Genes in Skeletal Morphogenesis, Differentiation, and Remodeling

During early development, *Dlx* genes are expressed by CNCCs, but not by mesodermal precursors or by postmigratory NCCs of the trunk. Analysis of the skull of mice carrying mutation or deletion of *Dlx1, Dlx2, Dlx3, Dlx5*, and *Dlx6* have shown that they regulate proximo-distal patterning of the pharyngeal arches and, in the case of *Dlx5/6*, determine maxillo-mandibular identity [42,43,53,54,55]. *Dlx* mutants do not present alterations in the shape of axial skeletal components, with the exception of the *Dlx5/6* double mutant which presents a limb malformation deriving from an early apical ectodermal ridge defect [53,56,57]. The limb and facial malformations observed in *Dlx5/6* double mutant mice are reminiscent of those observed in split hand foot malformation type I (SHFM1), a form of clinically and genetically heterogeneous human ectrodactyly [58]. The SHFM1 locus was mapped to 7q21.3 by Scherer et al. (1994) [58] using somatic cell hybrid lines from cytogenetically abnormal individuals, demonstrating that *DLX5* and *DLX6* are located within the SHFM1 critical interval and thus are candidate genes for this condition. The functional characterization of the *DLX5/6* regulatory region identified 26 tissue-specific enhancers [59] capable of directing the expression of *Dlx5/6* either in the limb, in craniofacial regions, and/or in the brain during development. Remarkably, while SHFM1 patients show mostly autosomal dominant inheritance with variable expressivity and reduced penetrance, in the *Dlx5/6* double mutant model, the ectrodactyly phenotype is mostly recessive with rare defects observed in heterozygous mice [53,56].

In general, during early development, *Dlx* genes play a major morphogenetic role in craniofacial bones while *Hox* genes determine the identity and shape of axial and appendicular bones. These early morphogenetic functions of *Dlx* genes are the result of highly dynamic exchanges of regulatory signals between the different cellular components of the developing embryo [43]. Later in development, *Dlx2, Dlx3, Dlx5*, and *Dlx6* assume new functions in different organs; in particular, in the post-natal skeleton these genes are involved in the control chondrogenesis and osteogenesis (see Figure 1).

Starting at mid gestation, *Dlx2, Dlx3, Dlx5*, and *Dlx6* are expressed in most sites of cartilage condensation and in differentiating osteoblasts in areas of bone formation [36,60,61,62]. As will be detailed later, the expression of *Dlx* genes persists then throughout life both in precursor bone marrow mesenchymal stem cells (BM-MSCs) and in differentiated chondrocytes and osteoblasts, suggesting their involvement in bone remodeling. 

The adult skeleton, and bone in particular, is continuously self-regenerating through a process called bone remodeling [63,64]. It has been estimated that in humans, most skeletal elements are renewed each decade thanks to the proliferation and differentiation of undifferentiated progenitor cells [64]. Although remodeling is similar for all bones, one should remember that the axial and appendicular skeletons have a different embryonic origin than that of the head. The cephalic skeleton derives exclusively from post-migratory cranial neural crest cells (CNCCs), whereas the rest of the skeleton derives from mesodermal precursors [65,66]. CNCC-derived tissues of the head do not express *Hox* genes, while axial skeletal elements are *Hox*-positive. Remarkably, progenitor cells assuring bone remodeling maintain the *Hox* positive/negative identity of their original tissue, even in the adult [67]. Apparently, therefore, there are distinct sub-populations of cephalic and axial skeletal stem cells.

In the following sections we will summarize the effects of *Dlx* genes in chondrogenesis and in osteogenesis. Their role in determining skeletal morphogenesis is not included in this review.

## 4. *Dlx* Genes in Cartilage Development and Maintenance

*Dlx* genes are involved in cartilage formation and maintenance; they play multiple roles in the recruitment of multipotent mesenchymal cells to the chondrogenic lineage and in their subsequent maturation to chondrocytes (Figure 1).

### 4.1. Dlx1 and Dlx2

Several in vitro studies suggest that *Dlx*2 might be directly involved in the regulation of chondrocyte differentiation. *Dlx*2 is expressed by the mouse clonal chondroblast cell line TMC23, and its expression levels increase during chondrocyte differentiation [68]. BMP2 treatment of these cells at early stages of chondroblast differentiation induces the expression of *Dlx*2; therefore, Dlx2 is a downstream target of the BMP2 signaling pathway. Interestingly, the same study indicated Col2alpha1 as a potential direct target of Dlx2. In summary, these authors propose a model in which BMP-2 stimulates *Dlx*2 expression, which is then a necessary transcription factor for *Col2alpha1* gene expression through a chondrocyte-specific enhancer [69].

The role of *Dlx2* on chondrocyte differentiation has been recently addressed by overexpressing *Dlx2* in the TCM23 cell line. Overexpression of *Dlx2* resulted in the accumulation of Aggrecan and type II collagen, two of the major components of the cartilage matrix. This accumulation does not seem to derive from enhanced transcription of the *Aggrecan* or *type II collagen* genes, but from a reduced expression of the protease *MMP13* induced by *Dlx2* overexpression. MMP13 plays a central role in extracellular matrix (ECM) degradation during late stages of chondrongenic differentiation, degrading both type II collagen and Aggrecan [70]. *Dlx*-response elements are present in the *MMP13* promoter, suggesting that Dlx2 might inhibit *MMP13* expression by directly binding to them [68].

Double inactivation of *Dlx1/2* in the mouse results in the generation of ectopic cartilages in the cephalic region [54] and alters the fate of an odontogenic cell population to become chondrogenic, as seen by the induced expression of *Sox9* [71]. These alterations might, however, be ascribed to changes in craniofacial patterning and not necessarily to a modified cell fate due to *Dlx1/2* deletion. Over expression of *Dlx2* in the mouse leads to postnatal condyle malformation, subchondral bone degradation, and irregular histological structure of the condylar cartilage. These mice present a profound alteration in the development of the post-natal condyle and might be considered animal models of temporal-mandibular joint osteoarthritis [72].

### 4.2. Dlx3

A role of Dlx3 in cartilage development and maintenance is suggested by its patterns of distribution but has not yet been analyzed by targeted mutations in vivo. At E15.5 of mouse development, *Dlx*3 is expressed in the growth plate of long bones by pre-hypertrophic and hypertrophic chondrocytes, but not in the resting, proliferative, and calcified cartilage zones [73]. In more mature limbs (E16) *Dlx*3 continues to be expressed by pre-hypertrophic chondrocytes, but its expression decreases in hypertrophic chondrocytes [73,74]. In primary cultures of rat nasal septum chondrocytes, *Dlx*3 is expressed at all stages of differentiation, from pre-cartilaginous condensations to differentiated chondrocytes. The expression pattern of *Dlx*3 parallels that of Indian hedgehog (*Ihh*) with the highest expression level observed in pre-hypertrophic chondrocytes, approximately one week before the onset of matrix mineralization. More recently, the expression of *Dlx3* in pre-hypertrophic and hypertrophic chondrocytes has been confirmed in post-natal mouse long bones up to five weeks after birth [75]. The increase in *Dlx*3 expression in chondrocytes leaving the proliferative stage and the maintenance of its expression in the post-natal growth plate suggest an important role of this protein in the regulation of growth plate dynamics [73].

### 4.3. Dlx5 and Dlx6

*Dlx*5 and *Dlx6* are expressed during chondrogenesis in mouse and avian embryos. *Dlx*5 is strongly expressed by condensing limb mesenchyme early during limb bud development [36,39,76,77,78]. In early chick and mouse embryos (up to E12.5 for the mouse), the expression of *Dlx*5 overlaps that of *Sox9* and type II collagen [78,79], suggesting that *Dlx*5 is an early marker of limb mesenchyme differentiation towards the chondrogenic lineage. Later, during axial chondrogenic condensation, the expression of *Dlx5/6* does not parallel that of *Sox9*. Starting at E13.5, *Dlx*5 is expressed in the mouse growth-plate in proliferating chondrocytes and in Indian hedgehog (*Ihh*)-positive pre-hypertrophic chondrocytes. At E14.5, *Dlx*5 is still expressed in the prehypertrophic zone and continues to be expressed in flattened chondrocytes and in the hypertrophic cartilage zone, even where *Ihh* is downregulated. *Dlx5* is also strongly expressed in the perichondrium and in the growth plate cartilage–bone interface. Later, *Dlx5* expression persists in proliferating chondrocytes and many *Dlx*5-positive cells are present in the prehypertrophic zone [79]. 

Comparison of *Dlx*5^+/+^ and *Dlx*5^−^^/^^−^ E18.5 growth plates shows that, in the absence of the gene, the proliferating zone is significantly larger [79] while the hypertrophic zone is narrower. These data suggest that Dlx5 acts as a regulator of chondrocyte differentiation and has a smaller effect on the control of proliferation.

Overexpression of *Dlx5* in ovo with an avian replication-competent retroviral vector (RCAS-*Dlx*5) resulted in a clear reduction in the proliferative zone with an increase in the size of the hypertrophic zone. In *Dlx*5-infected chicks, there was a higher number of large, round cells in the hypertrophic zone [80]. Therefore, *Dlx*5 overexpression in the chicken induces a cartilage phenotype opposite to the one observed in *Dlx*5^−^^/^^−^ mutant mice. 

Several in vitro experiments suggest that Dlx5 is an important regulator of multipotent mesenchymal cell recruitment to the chondroblastic lineage and plays a role in chondrocyte maturation. *Dlx*5 has an anti-proliferative effect on chondrocytes and fibroblasts. Micro-mass assays using chicken embryo limb bud mesenchyme infected with RCAS-*Dlx*5 show an increased chondrogenic differentiation compared to a non-recombinant RCAS-infected mesenchyme. It appears, therefore, that Dlx5 regulates the progressive transition of immature proliferating chondrocytes to pre-hypertrophic and hypertrophic chondrocytes, confirming the notion that Dlx5 acts as a positive regulator of chondrogenic differentiation. During development, *Dlx5* and *Dlx6* are often co-expressed and might share partially redundant functions. Simultaneous disruption of *Dlx*5 and *Dlx*6 in the mouse results in a delay of the normal chondrocyte developmental program [53]. At developmental stages in which pre-hypertrophic, hypertrophic, and calcified chondrocytes were observed in *Dlx**5/6^+/+^* and *Dlx5^+/−^* mice, only pre-hypertrophic chondrocytes were present in *Dlx**5/6*^−^^/^^−^ mutants. Furthermore, in *Dlx*5/6^−^^/^^−^ embryos, Col2a1 and Col10a1, markers of pre-hypertrophic and hypertrophic chondrocytes, respectively, were still expressed in articular sections, whereas in *Dlx*5/6^+/−^ embryos, both genes had already been significantly downregulated indicating a severe delay in cartilage differentiation. Both the Fgf/Fgfr3 and the Ihh/Pthrp/Pthrp-R signaling pathways are important for the regulation of chondrocyte proliferation and differentiation, and for the coupling of chondrogenesis to osteogenesis during endochondral ossification [81,82]. In line with what has been observed for Col2a1 and Col10a1, there is a delay in *Fgfr3* and *Ihh* expression in *Dlx*5/6^−^^/^^−^ developing cartilages, resulting in contrasting patterns of *Fgfr3* and *Ihh* expression in control and *Dlx*5/6^−^^/^^−^ embryos, while *Pthrp-R* expression is unaffected [53]. Therefore, the expression of most key regulators for chondrogenesis is severely delayed in *Dlx*5/6^−^^/^^−^ developing skeletons, resulting in a severe disruption in the differentiation of chondrocytes to *Runx2*-expressing cells.

*Dlx*6 seems to have a redundant role with *Dlx5* during chondrogenesis: (1) low levels of *Dlx6* are expressed in the same territories as *Dlx*5 and *Sox9* during chicken limb bud formation, and (2) in micro-mass cultures, misexpression of *Dlx*6 stimulates the differentiation to chondroblasts as *Dlx*5. It appears, however, that Dlx5 and Dlx6 utilize different domains to regulate chondrocyte differentiation. The *Dlx*5 N-terminal domain and the homeodomain are required to stimulate mesenchymal differentiation along the chondrogenic pathway, whereas all the domains of Dlx6 are needed for full chondrogenic activity [78].

### 4.4. Implication of Dlx5 in Osteoarthritis

Osteoarthritis (OA) is one of the major causes of disability [83]. The most frequent risk factors for OA include age, gender, prior joint injury, obesity, genetic predisposition, and mechanical factors [84,85]. Despite the multifactorial origin of OA, the pathological lesions seen in osteoarthritic joints present similar features, resulting in pain, deformity, and loss of function. OA can be seen as a symptom-complex characteristic of several genetically programmed and/or behaviourally induced disorders. In adults, OA often results from biomechanical forces after joint impact that, by acting continuously on articular cartilages, can generate focal injuries that can commonly progress into chronic degeneration [86]. Cartilage tissue is not vascularized and has very limited regenerative potential [87]. Thus, cartilage injuries are associated with an increased risk for developing OA [88]. Cell hypertrophy is a hallmark of OA. Cartilage samples derived from patients presenting different OA statuses demonstrated a positive correlation between the levels of *DLX5* expression and the severity of osteoarthritis, reinforcing the notion that DLX5 is a regulator of cellular hypertrophy, and that its inhibition could improve cell-mediated cartilage tissue repair [89]. Indeed, *DLX5* expression increases with chondrocyte differentiation [90] and in OA; cultured chondrocytes from OA patients express higher levels of *DLX5* and *COL10*, compared with non-arthritic chondrocytes. DLX5 inhibition in BM-MSCs was found to reduce cellular hypertrophy without inhibiting chondrogenic potential, while overexpression of DLX5 was associated with increased expression of cellular hypertrophy markers [89]. Articular cartilage calcification indicates cartilage damage. The pharmacological inhibition of DLX5 by administration of DLX5-blocking antibodies reduced the cartilage calcification associated with osteoarthritis in rabbits along with the gene expression of alkaline phosphatase [91]. Following this initial observation, Lu et al. recently reported that anti-Dlx5 treatment improved papain-induced osteoarthritis in a rabbit model. Indeed, chondrocyte hypertrophy and apoptosis, and extracellular matrix damage, which were increased in damaged cartilage, were all reduced after anti-Dlx5 treatment [92]. It must be noted, however, that in these rabbit models the lesion was induced artificially through repeated intra-articular injections of papain; this is profoundly different from what occurs in genetically-determined and acquired bone disorders with associated cartilage damage.

## 5. Roles of *Dlx* Genes in Osteoblast Differentiation

### 5.1. Dlx1 and Dlx2

*Dlx1* and *Dlx2* play an important role in tooth development [71,93] and in dermatocranial bone patterning [42]. However, to our knowledge, no study has shown the presence of these proteins in the osteoblastic cell lineage. Since this review does not cover tooth morphogenesis and craniofacial patterning, these aspects will not be further analyzed.

### 5.2. Dlx3 and Dlx4

During endochondral ossification, *Dlx*3 is expressed in cells of the osteoblastic lineage and in the inner layer of the periosteum. In E15.5 mouse embryos, *Dlx*3 is present in osteoblast progenitors, in osteoblasts lining bone trabeculae and in osteocytes at early stages of differentiation, whereas mature osteocytes do not express this gene. At E16.5, *Dlx*3 is still expressed by cells surrounding the trabeculae and by endosteal osteoblasts of the diaphyseal cortical bone [74]. In primary rat osteoblast culture, the highest level of *Dlx*3 mRNA and protein is present in induced mature osteoblasts, as seen by the concomitant upregulation of *osteocalcin* and *Runx2* and downregulation of *Msx2*. Treatment of these cells with BMP2 results in a 10-fold increase in *Dlx*3 expression, suggesting that *Dlx*3 is an early BMP2-induced gene during bone formation [73,74].

Overexpression of *Dlx*3 in MC3T3-E1 cells induced an increased expression of the osteoblast-specific genes *Collagen type1*, *Bone sialo protein*, *Osteopontin*, and *Osteocalcin*. ChIP analysis provided evidence of direct binding of Dlx3 onto the *osteocalcin* regulatory region [74].

Tricho–dento–osseous syndrome (TDO) is an autosomal dominant disorder characterized by enamel hypoplasia, severe taurodontism, moderately increased trabecular bone mineral density, and typical kinky/curly hairs. TDO is linked to chromosome 17q21 markers with no indication of genetic heterogeneity [94,95]. The TDO locus has been mapped to the chromosomal region that includes the *DLX3/4* tandem and, more specifically to a 4bp deletion in the human *DLX*3 gene (*DLX*3 4bp DEL, NT3198 mutation; OMIM 600525) [96,97,98]. Aside from taurodontism and enamel hypoplasia, TDO patients carrying this 4bp deletion often present significantly higher bone mineral density in the radius, ulna, spine, and hips compared to unaffected family members [99]. The 4bp *DLX3* deletion results in a frameshift that changes the last 97 C-terminal amino acids and gives rise to a novel 119 amino acid C-terminal peptide in the mouse *Dlx3* cDNA, just 3′ to the homeobox. The homeodomain region in both human and mouse *DLX3* genes includes a nuclear localization signal (NLS) [100]. The 4bp *DLX3* deletion does not alter the structure of the homeobox nor of the NLS regions; therefore, the nuclear translocation ability of the mutant DLX3 protein is unchanged.

The trabecular bone volume and the mineral density of mice carrying the *Dlx3* 4bp deletion is increased both in young and adult individuals. As the rate of bone formation does not increase in vivo in these mice, their phenotype was suggested to derive from decreased osteoclast formation and bone resorption due to the increased serum levels of IFN-*γ* [101].

To address the in vivo role of DLX3 in the regulation of osteoblastogenesis, bone density, and remodeling in the appendicular skeleton, Isaac et al. [75] generated mice carrying the conditional inactivation of *Dlx3* either in mesenchymal or in osteogenic lineage cells. Both models are characterized by an increase in bone mass throughout their lifespan. In the absence of *Dlx3*, endochondral bone formation is significantly higher, with the trabecular bone extending into the medullary cavity while the cortical bone is thicker and more porous, presenting a reduced bone mineral density. Combining in vivo gene profiling and ex vivo cellular analyses, the authors concluded that increased trabecular bone mass in the Dlx3-deleted mice depends on enhanced osteoblast bone-forming activity. Indeed, RNA-Seq analysis of *Dlx3*-deleted metaphysis shows upregulation of transcription factors associated with osteoblastogenesis, including *Runx2*, *Sp7*, and *Dlx5/Dlx6* [60,75]. 

The notion that Dlx3 regulates osteoblast activity is reinforced by the fact that *Dlx3*-deleted bone marrow stroma cells (BMSCs) also show increased osteoblastic differentiation with increased expression of *Runx2* and *Dlx5*. Collectively, these observations suggest that DLX3 is a negative regulator of osteoblastogenesis. This notion is further supported by a ChIP analysis on BMSCs, which shows that DLX3 binds to the promoters of *Sp7*, *Dlx5* and *Dlx6*, and *Runx2*, directly modulating their activity (see also [102]).

*DLX4* is expressed at very low levels from most adult tissues, including bone, and its potential role in skeletogenesis has not been studied directly. *DLX4* is, however, expressed at high levels in many types of tumors, including in leukemia, lung, breast, ovarian, and prostate cancers suggesting an implication in the control of cell proliferation [103,104,105,106,107,108]. A role of *Dlx4* in craniofacial skeletal morphogenesis has been suggested by its expression in the mesenchyme of murine palatal shelves during embryonic development, and by the fact that a specific mutation of *DLX4* (c.546delG) is associated with familiar cleft lip and/or palate [109].

### 5.3. Dlx5 and Dlx6

*Dlx*5 is expressed by osteoblasts from very early stages of bone development [36,76,110] and persists in these cells throughout life [111]. In perinatal and post-natal bone, *Dlx5* expression is predominantly found at the periphery of diaphyseal bone and in cells surrounded by osteoid within the trabeculae [62,112]. 

*Dlx*5^−^^/^^−^ mice die at birth. Histopathological analysis of their long bones performed in late embryos showed that both trabecular and cortical bone components were affected. By E15.5, *Dlx*5^−/−^ mice present a significant reduction in the ossified portion of long bones. At birth, histological analysis revealed a lesion characterized by the presence of a more complex structure of the endosteal component of the diaphysis, which forms an elaborate mesh of woven bone, and by a reduction in the periosteal bone lamina [62]. Delayed ossification of the parietal, interparietal, and superoccipital bones of *Dlx*5^−/−^ mice results in open fontanelle. These morphological findings, both in endochondral and intramembranous bone types, suggest that the absence of *Dlx*5 results in a generalized defect in osteogenesis. 

Indeed, the in vitro analysis of *Dlx5*^−/−^ cells suggests that this transcription factor promotes osteoblast proliferation and differentiation as indicated by the decreased expression of bone differentiation markers and their reduced capacity to generate mineralized nodules in vitro [112]. These *Dlx5*^−/−^ osteoblastic defects may depend on *Runx2*-dependent or -independent pathways. As will be discussed later, *Dlx5* acts as a transcriptional activator of *Runx2* in bone by binding to its P1 promoter [113] and to a *Runx2* bone specific enhancer; consequently, *Dlx5* inactivation results in decreased *Runx2* expression. *Dlx5* can also activate osteoblast-specific genes, such as *ALP* and *osteocalcin*, in the absence of *Runx2*, suggesting a *Runx2*-independent pathway [102]. *Osx* expression is also reduced in cultured *Dlx5*^−/−^ osteoblasts, possibly through a Dlx5-dependent/Runx2-independent mechanism [114]. Both *osteocalcin* and *BSP*, markers of osteoblast differentiation, are drastically downregulated in *Dlx5*^−/−^ osteoblasts, as predicted by the analysis of their regulatory regions and by in vitro studies [110,115]. Remarkably, no significant difference in trabecular thickness, an indicative parameter of osteoblastic activity, was observed in *Dlx5*^−/−^ mice at birth. This apparent contradiction may depend on the different origin of cortical and trabecular osteoblast precursors. Indeed, cultured calvaria-derived cells are more similar to cortical than to trabecular osteoblasts from long bones [112].

Simultaneous inactivation of *Dlx*5 and *Dlx*6 results in more pronounced abnormalities of endochondral bones than the single inactivation of *Dlx*5, suggesting a redundant function for the two genes. Serial skeletal sections showed the presence of vascularization and mineralized bone matrices in heterozygous *Dlx*5/6^+/−^ embryos, while comparable regions in *Dlx*5/6^−/−^ embryos consisted of hypertrophic and calcified chondrocytes, with minimal vascular invasion and a predominantly cartilaginous matrix. Molecular analysis of comparable E16.5 *Dlx*5/6^−/−^ embryos also suggested a delay in the onset of the osteogenic pathway. Similar *Runx2* expression patterns were observed within the chondrium and perichondrium of *Dlx*5/6^+/−^ and *Dlx*5/6^−/−^ mice, while *osteocalcin* gene expression was dramatically reduced in *Dlx*5/6^−/−^ skeletal sections. Altogether, these results suggest that *Dlx*5/6 have a partially redundant positive role in the osteoblast maturation pathway, and that, in their absence, the accumulation of mature osteoblasts is severely retarded or absent [53].

### 5.4. Association between DLX5/6 and Human Bone Mineral Density

DLX5/6 are associated with a type of human ectrodactyly known as split hand foot malformation type I (SHFM1) [116]. The inactivation of *Dlx5/6* in the mouse results in a similar limb phenotype, reinforcing the notion that these genes are responsible for this disease [53,56]. The characterization of the *DLX5/6* regulatory region performed by generating reporter transgenic animals has permitted the identification of 26 tissue-specific enhancers (eDlx#XX) [59] capable of directing the expression of the genes either in the limb, in craniofacial regions, and/or in the brain during development. However, the role of these enhancers in the adult is unknown.

Remarkably, at least five GWAS studies have found a strong association between adult bone mineral density and SNPs located in the 7q21.3 region, including eDlx#18 [117,118,119,120,121]. Furthermore, a haplotype controlling bone mineral density and osteoporosis susceptibility has been reported in this locus [122]. This haplotype is constituted by a succession of co-segregating SNPs located in intronic regions close to the gene *SEM1/FLJ4220/DSS1*, which is associated to SHFM1 but has no known function. This locus includes upstream enhancers of *DLX5/6*. Collectively, these observations prompt us to conclude that polymorphisms in SNPs regulating *DLX5/6* expression are involved in BMD determination and might be considered susceptibility factors for osteoporosis (Figure 2).

## 6. *Dlx* Genes as Modulators of Osteoclast Activity

Osteoclasts are large, multinucleated cells derived from the monocyte–macrophage lineage of hematopoietic cells [123,124] responsible for bone matrix resorption. The main biological roles of osteoclasts are to assure skeletal remodeling, to maintain serum calcium levels, and to participate in bone fracture healing [125].

In several human diseases, such as malignant hypercalcemia or postmenopausal osteoporosis, an increase in bone resorption is the key pathophysiological event. In contrast, several rare disorders derive from decreased resorption leading to osteopetrosis.

To maintain a normal bone mineral density, it is important that the activity of osteoblasts and osteoclasts is well coordinated. Osteoporosis is primarily caused by the unbalanced activity of osteoclasts and osteoblasts. Dlx gene expression analysis in cultured osteoclasts has shown that: (1) *Dlx3/4* are never expressed by these cells; (2) *Dlx5/6* are expressed at low level, but their presence is not directly associated with osteoclast differentiation and, (3) *Dlx1/2* are present in osteoclasts and might be modulators of their rate of differentiation without having a direct role in osteoclastogenesis [126].

Although *Dlx* expression by osteoclasts does not seem to be directly responsible for osteoclastogenesis, these genes seem to play an important indirect role in the control of osteoclastogenesis and/or osteoclasts activity.

In addition to impairment of calvaria-derived osteoblast function, *Dlx5*^−^^/^^−^ mice showed a significant increase in osteoclast number and trabecular separation, suggesting a higher level of bone resorption, suggesting that *Dlx5*^−/−^ osteoblasts indirectly enhanced osteoclastogenesis [111]. As *Dlx5* is not expressed by differentiated multinucleated TRAP-positive osteoclasts, this finding suggests that *Dlx5*^−/−^ osteoblasts could alter the crosstalk between osteoblasts/osteoclasts and indirectly enhance osteoclastogenesis by acting on the molecular triad OPG/RANK/RANKL, which orchestrates osteoclastogenesis and bone resorption [127]. Indeed, *Dlx5*^−/−^ osteoblasts presented an increased RANKL/OPG ratio and *Dlx5*^−/−^ co-cultures displayed a higher number of multinuclear TRAP positive cells with a higher resorption activity, indicating a higher number of functional osteoclasts. The relatively undifferentiated status of *Dlx5*^−/−^ osteoblasts might favor osteoclastogenesis as previously observed [128]. As the osteoclast number and the RANK-L/OPG ratio are decreased in *Runx2*^−/−^ mice, it seems that Dlx5 acts on osteoclastogenesis independently from Runx2 [129]. 

The osteoblast/osteoclast coupling associated with increased resorption activity in *Dlx5*^−/−^ mutants could permit a better understanding of the origin of bone homeostasis-related diseases, such as osteoporosis or osteopenia, resulting from immobilization. Indeed, it has been shown that *Dlx5* expression increases in the presence of mechanical loading thus altering osteoblasts/osteoclasts coupling [130,131].

The analysis of other mouse models, such as mice carrying conditional alleles of *Dlx5* or *Dlx5/6* in bone, will be decisive to evaluate the potential implication of these genes in osteoporosis.

An interesting concept that emerges from the analysis of the *Dlx3* and *Dlx5/6* mutant phenotypes is that mutual regulation of these two genes constitutes a regulatory network essential for the maintenance of bone homeostasis. First of all, Dlx3 is an inhibitor of *Dlx5/6* in bone, whereas previous studies showed that the targeted simultaneous inactivation of *Dlx5* and *Dlx6* in the embryo results in the loss of *Dlx3* expression [132]. These finding suggest that a reciprocal regulatory loop between Dlx5/Dlx6 and Dlx3 might be present during bone formation and homeostasis. Both Dlx3 and Dlx5 bind to the *Ocn* promoter at the onset of transcriptional activation; later, during mineralization, Dlx3 occupancy decreases sharply, while Dlx5 occupancy remains maximal [102]. In line with these findings, *Dlx3* deletion in calvaria osteoblasts results in increased occupancy of Dlx5 and premature occupancy of Runx2 on regulatory elements on the *Ocn* promoter [75]. Thus, Dlx3 and Dlx5 might have coordinated and opposite roles on the regulatory network supporting osteoblastic differentiation and bone-formation acting.

As mentioned before, *Dlx3* inactivation results in increased cortical bone thickness and higher mineral apposition rate in the periosteum, decreased BMD with increased porosity, and vascularization. RNA-Seq analysis of cortical bone of *Dlx3*^−/−^ mice revealed upregulation of *Opg* and *Mepe*, two major inhibitors of osteoclastogenesis expressed by late-differentiated osteoblasts and osteocytes. These findings suggest that in normal conditions, *Dlx3* is a suppressor of *Opg* and *Mepe* in cortical bone, favoring osteoclastogenesis and bone resorption. It seems plausible that the mutual and opposite regulatory effects of Dlx5 and Dlx3 might play an important role in osteoblast/osteoclast coupling and maintenance of bone homeostasis. Conversely to Dlx5, Dlx3 would inhibit osteoblast bone-forming activity through the negative transcriptional control of bone formation genes, while simultaneously activating bone resorption through osteoblast-activated regulation of osteoclastogenesis. 

## 7. Regulatory Cascades Involving *Dlx* Genes during Skeletal Formation

### 7.1. Interaction between BMPs and Dlx

Bone morphogenetic proteins (BMPs), members of the TGFß superfamily, are activators of the osteogenic program [133]. BMPs stimulate the differentiation of mesenchymal and osteoprogenitor cells towards the osteoblastic lineage and control their apoptosis [134,135]. The expression of *Dlx* family members is rapidly induced in response to BMP-mediated osteoblast differentiation.

Although MC3T3-E1 cells transfected with *Dlx*5 are able to undergo differentiation and even mineralization in absence of BMP [136], BMP4 induces *Dlx*5 expression in osteoblastic MC3T3-E1 cells, in mouse embryos, and in fractured bones of adult mice. Similarly, induction of *Dlx*5 by BMPs has been observed also during early avian skull development [137]. BMP-2 stimulates the recruitment of Dlx5 and to a lesser extent of Dlx3 to the *osteoactivin/GPNMB* promoter region [138] and induction of *Dlx3* and *Dlx5* after BMP-2 treatment coincides with commitment of pluripotent C2C12 mesenchymal cells to the osteogenic lineage [139].

It has been shown that *Dlx5* expression in osteogenic cells is specifically induced by BMP-2 or -4 signaling but not by other osteotrophic signals or other TGF-β superfamily members. The notion that *Dlx*5 is a target of the BMP signaling pathway is also supported by the fact that *Dlx*5 transcription is not only stimulated by BMP2 treatment, but also by the overexpression of constitutively active members of the BMP pathway: BMP-receptor-IA and IB, Smad1, and Smad5 [114]. Conversely, in vivo studies of *Dlx*5^−/−^ mice suggest that BMP2/4 could in turn be targets of *Dlx*5. For example, during inner ear morphogenesis, *BMP2/4* expression is strongly downregulated in *Dlx*5^−/−^ embryos [140].

### 7.2. Contribution of Dlx5 in Regulating the Expression of Runx2

As mentioned above, *Runx2* is a key regulator of osteogenesis that directs multipotent mesenchymal cells towards the osteoblastic lineage. Mutations of *RUNX2* lead to cleidocranial dysplasia, and heterozygous loss-of-function alleles in the mouse also lead to a similar phenotype [5]. Osterix (Osx, Sp7), a second key transcriptional regulator of osteogenesis [3], is expressed in all developing bones and plays a critical role in bone formation. Runx2 controls the progression of differentiation from the commitment step to the point at which osteo-chondro progenitor cells appear. In contrast, Osx acts mainly during the terminal differentiation of osteoblasts and distinguishes the osteogenic from the chondrogenic pathways. Both *Runx2* [141,142] and *Osx* [3] are activated by BMP2 treatment. However, as pre-treatment with cycloheximide (a de novo protein synthesis inhibitor) blocks the BMP2-induced expression of *Runx2* [114] and *Osx* [143], these osteogenic master genes do not seem to be direct targets of the BMP-signaling cascade but require the intermediation of newly synthesized proteins. In contrast to *Runx2* or *Osx*, *Dlx*5 induction is unaffected by cycloheximide pre-treatment [114,141] indicating that *Dlx5* is a direct target of BMP signaling and may act as an upstream regulator of *Runx2* and *Osx* in the BMP2 signaling pathway.

This hypothesis is further supported by the fact that the inhibition of *Dlx*5 in cultured osteoblasts using antisense techniques results in complete inhibition of *Runx2* and *Osx* expression [144]. Moreover, *Dlx*5 overexpression is sufficient to induce *Runx2* expression in cultured cells, even in the absence of BMP2 treatment.

Chromosomal analysis has shown that, both in human and in mice, the *Runx2* gene has two distinct promoters that give rise to two different isoforms, *Runx2-I* and *Runx2-II.* The *Runx2-I* and *Runx2-II* isoforms are differentially expressed during intramembranous and endochondral bone formation [145,146]. *Runx**2-II* expression co-localizes with that of *Bmp*2 and *Dlx*5 during mouse calvaria bone development. Dlx5 specifically regulates the *Runx2-P1* promoter, but not the *Runx2-P2* promoter [113]. Three putative homeodomain response elements are present in the *Runx2-P1* promoter (see Figure 3) and deletion analysis shows the importance of these sites to induce a strong increase in reporter activity after *Dlx5* transfection in both non-osteogenic (C2C12) and osteogenic (ROS 17/2.8) cells [113]. Consistent with these findings, *Runx2-II* mRNA expression is stimulated by *Dlx*5 overexpression and inhibited by *Dlx5* antisense treatment, while *Runx**2-I* mRNA levels are unaffected. The differential regulation of the two promoters by *Dlx*5 correlates with their distinct temporal-spatial expression patterns during bone development [145,146,147,148]. 

More recently, a more detailed analysis of the Runx2 regulation has shown that a 6.1-kb region including the Runx2-P1 promoter is not sufficient to direct gene expression to osteoblasts in transgenic mice [149]. This finding has suggested that a regulatory element might be present upstream of the 6.1-kb P1 promoter. Indeed, an evolutionary conserved 1.3-kb region, located about 30 kb upstream of the transcription initiation site of *Runx2-II*, showed the presence of an osteoblast-specific enhancer. Analysis of this region, which presents histone modifications typical for enhancers, showed that it contains regulatory elements sufficient to direct *Runx2* expression to osteoblasts, and that Dlx5 and Mef2, which formed an enhanceosome with Tcf7, Ctnnb1, Sp7, Smad1, and Sox6, played a central role in the osteoblast-specific activation of the enhancer. It appears, therefore, that Dlx5 can act both on the P1 promoter and on the osteoblast-specific enhancer to direct a specific high-level *Runx2* expression during bone differentiation [149].

### 7.3. A Direct sp7/dlx5 Interaction in the Regulation of Bone Gene Expression

Recently, Hojo et al. [150] provided surprising evidence for a specific mode of Sp7/Osx action different from that of other members of the Sp transcription factors. Whereas most Sp family members bind a consensus GC-box sequence [151], the primary site of engagement of Sp7 in bone extract ChIPs was an AT-rich motif similar to homeodomain-response elements. Indeed, it was shown that Sp7 does not bind directly to DNA, but instead binds to Dlx family members directly associated with AT-rich target sequences. This led to a model in which Dlx members bind through their homeodomain to the core AT-rich motif, and in a second step Sp7 binds to Dlx factors through its modified zinc fingers. As a consequence of the fact that the action of Sp7 is exerted through its binding to Dlx5, the Sp7-associated genome in MC3T3E1-derived osteoblasts is almost entirely included within Dlx5-bound target regions. In essence, these findings support a model in which Dlx5 recruitment of Sp7 to osteoblast enhancers is instrumental for Sp7-directed osteoblast specification. Interestingly, comparative analysis of Sp7 sequences identifies a specific zinc-finger variant in all major vertebrate groups, but not in non-vertebrates, and raises the possibility that the emergence of an Sp7/Dlx5 interaction might have been central for the evolution of bone formation in vertebrates [150].

### 7.4. Dlx5/Runx2 Interplay in the Regulation of Osteoblast Differentiation

Several findings suggest that Dlx and Runx2 proteins may cooperate in the regulation of other osteoblast-specific genes. As stated above, Dlx3 interacts with Runx2 [74] and a similar type of protein–protein interaction has been suggested for the Dlx5/Runx2 interaction in the control of bone sialoprotein expression [152]. Both Dlx5 and Runx2 proteins can bind to the promoters of alkaline phosphatase and osteocalcin and stimulate their expressions with an additive effect. However, in Runx2^−/−^ cells, Dlx5 can still stimulate alkaline phosphatase expression in a Runx2-independent manner [153]. Among the four homeodomain core-binding sequences (ATTA or TAAT) present in the osteocalcin promoter, one is responsive to Dlx5. The role of Dlx5 in the transcriptional control of osteocalcin gene remains controversial, with certain studies showing that Dx5 enhances osteocalcin expression [136,154], while other studies show an inhibitory activity of Dlx5 [110]. 

Bone sialoprotein (BSP), a terminal marker of osteoblastic differentiation, appears to be an important direct target of Dlx5 and Runx2 in osteoblasts [152]. BSP is one of the major structural proteins of the bone-mineralized matrix. BSP also plays an important role in the differentiation of preosteoclastic cells into mature osteoclasts. Dlx5 binds directly to a conserved homeobox-binding site in the murine and human BSP promoters and stimulates its transcription [155,156,157]. The Dlx5-dependent regulation of BSP has also been observed in breast cancer cell cultures [158].

### 7.5. Antagonistic Action of Dlx5 and Msx2 in the Control of Bone Specific Gene Expression

The *Msx* gene family encodes homeobox transcription factors which display opposite transcriptional properties compared to *Dlx* genes. Dlx5 and Msx2 proteins seem to have antagonistic effects on osteoblast differentiation. While *Dlx5* stimulates osteoblast differentiation and activates the promoters of bone marker genes [153,155,159], Msx2 seems to act as a repressor of these genes and as a suppressor of osteogenic differentiation [160]. Several models have been suggested for the antagonistic action of Dlx5 and Msx2. The two proteins might directly interact which each other to form a functionally inactive complex where the DNA-binding homeodomains are inactivated [74]. Alternatively, the homeodomains of the Dlx5 and Msx2 proteins may compete for binding to common response elements in bone-specific marker genes, such as osteocalcin or the Runx2-P1 promoter. 

## 8. Possible Involvement of Dlx5 in Osteosarcoma

Osteosarcoma (OS) is a very common malignant bone cancer with high morbidity in young patients [161]. Although the introduction of neoadjuvant chemotherapies has permitted to extend the survival rate of OS patients, OS has a high risk of local recurrence and lung metastasis. DLX5 can act as an oncogene in lymphomas and lung cancers, possibly by controlling the expression of *MYC* [162]. Genome-wide genetic and epigenetic profiling of 19 human OS cell lines showed increased expression of DLX5 and RUNX2 [163]. DLX5 inactivation by siRNA in HOS and MG-63 OS cell lines inhibits cell growth and osteosarcoma progression, inducing apoptosis an both in vitro and in vivo. These effects were rescued by overexpression of NOTCH1 [164], which appears to be a direct target of DLX5. These findings support the notion that DLX5 might play an oncogenic role in OS through the activation of the NOTCH signaling pathway.

## 9. Concluding Remark

*Dlx* genes encode for transcription factors that play pivotal roles in chondrocyte and osteoblast differentiation and in bone remodeling in adults. However, the exact mechanisms involved in these regulations are not yet fully elucidated. To improve our understanding of the involvement of *Dlx* genes in skeletogenesis and bone homeostasis, it is essential to include in our conceptual horizon further levels of regulation, such as DNA-looping [49], methylation, imprinting [47], *ncRNAs* [50], and *miRNA*s. All these regulatory modes have been shown to act on the expression of *Dlx* genes during development and might, therefore, be also involved in the maintenance of the adult skeleton. The inclusion of these advanced levels of regulation might provide new hints to address skeletal disorders, such as osteopenia or osteoporosis.

## Figures and Tables

**Figure 1 cells-11-03277-f001:**
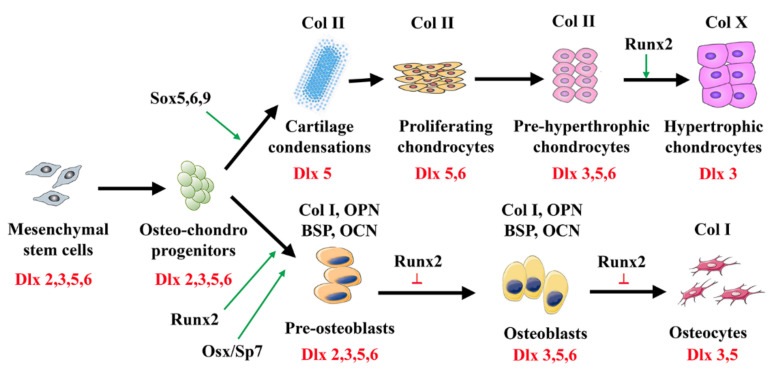
Role of Dlx transcription factors in the control of cartilage and bone differentiation. Osteoblasts and chondrocytes differentiate from common mesenchymal skeletal stem cells. *Dlx* genes are essential for the commitment of progenitors to both the chondrocytic and osteoblastic lineage and for the differentiation of both osteoblasts and chondroblasts. Abbreviations: Col-I/II/X, type I/II/X collagen; BSP, bone sialoprotein; OPN, osteopontin; OCN, osteocalcin.

**Figure 2 cells-11-03277-f002:**
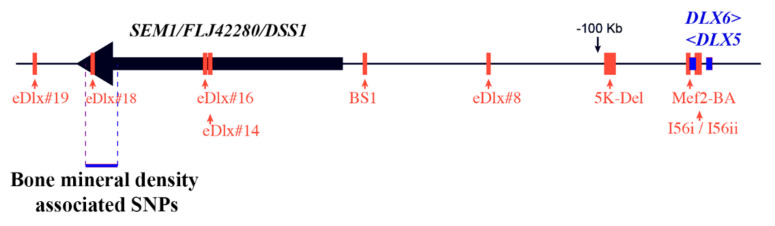
−A haplotype controlling bone mineral density spans over a *DLX5/6* expression enhancer. The functional characterization of *DLX5/6* regulatory region identified several enhancers (indicated in orange in the figure) directing the expression *DLX5/6* [59]; furthermore a haplotype controlling bone mineral density and osteoporosis susceptibility has been reported in this locus [122]. At least four GWAS studies have found a strong association between bone mineral density (BMD) and SNPs in the 7q21.3 region, which includes the enhancer “eDlx#18” [117,118,119,120,121], prompting our suggestion that DLX5/6 should be included in the list of genes controlling BMD. This region is located in proximity to the gene *SEM1/FLJ4220/DSS1*, of unknown function, but associated to SHFM1.

**Figure 3 cells-11-03277-f003:**
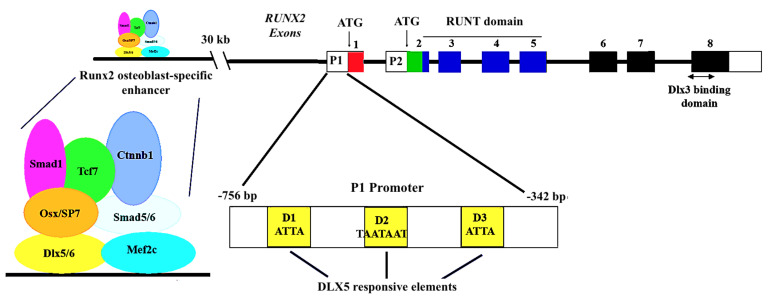
Genomic organization of the mouse Runx2/Cbfa1 gene locus. *RUNX2* can be activated by two different promoters (P1 and P2) that give rise to two isoforms of the protein. The P1 enhancer which drives bone-specific *RUNX2* expression includes three DLX5 responsive elements (D1, D2, D3). P1 alone is not sufficient to drive the expression of *RUNX2* in bone, but requires an enhancer located about 30kb upstream; this enhancer binds DLX5/6 and Mef2C and other co-activator factors, such as SP7, that interact directly with DLX5. DLX5 is therefore needed both at the promoter and enhancer level for the activation of *RUNX2* in bone.

## Data Availability

No new data published, all data cited are available in the press.

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
