# Peer review of "DLX Genes in the Development and Maintenance of the Vertebrate Skeleton: Implications for Human Pathologies"

_cells, 2022, doi:10.3390/cells11203277_

Round 1

Reviewer 1 Report

Nicely written Review article. Authors need to strengthen the clinical part.

Line 190-191: 

Comment: SHFM I, is a condition which is clinically and genetically heterogeneous and shows mostly autosomal dominant inheritance with variable expressivity and reduced penetrance. DLX5, DLX6 deletion causing effectively, autosomal dominant ectrodactyly (split hand, aplasia of single digital ray, hypoplasia, and triphalangeal thumb). Moreover, lower limbs with broad hallux and clinodactyly are involved in the pathogenetic mechanism.

Line 328-332: Authors wrote, Osteoarthritis (OA) is one of the major causes of disability. It results from biomechanical forces after joint impact that, by acting continuously on articular cartilages, can generate focal injuries that can commonly progress to chronic degeneration.

Comment: Authors restricted the etiology of OA in connection with biomechanical forces i.e post-adulthood pathology. In clinical practice, the genetically programmed disorders have the upper hand. OA is mostly manifest itself as part of a multisystem disorder. In other words, OA is a symptom complex rather than a diagnostic entity until prove otherwise. The vast majority of medications administered to patients in departments of rheumatology are proved of being hepatotoxic.

Line 338-342, authors wrote; Cartilage calcifications, is one of the hallmarks of osteoarthritis, are at the initiation of cartilage damage. The pharmacological inhibition by DLX5 blocking antibodies reduced the cartilage calcification associated with osteoarthritis in rabbits along with the gene expression of alkaline phosphatase.

Comment: Calcified cartilage deposited during endochondral bone formation particularly in the long list of genetically determined and acquired bone sclerosing disorders, such osteopetrosis, Camurati-Engelmann Disease, endosteal hyperostosis, craniodiaphyseal dysplasia, osteopoikilosis, osteopathia striata, pachydermoperiostosis, hepatitis C-associated osteosclerosis and other high bone mass phenotype. These disorders can never respond to the pharmacological inhibition through administering DLX5 blocking antibodies. On the contrary, these medications can lead to liver damage.

Line 367-379;

Comment: The clinical phenotype of DLX3/4 deletion in patients with TDO is mainly taurodontism and enamel hypoplasia.  The other phenotypes experienced in TDO such as nail defects and sclerosing bone disorders are variable. 

Reviewer 2 Report

Summary:

This review aims to summarize the major findings concerning the involvement of Dlx genes in skeletal development and homeostasis and their involvement in skeletal aging and pathology. In section 1 the authors give a broad molecular background into skeletal development and maintenance this is very detailed and encapsulates a large body of work. The authors manage to cover all relevant aspects well to set the scene for further descriptions. Authors provide a very thorough description of Dlx genes in chondrogenesis, both in vitro and in vivo data and subsequent description in osteoarthritis models and in patient cell culture experiments. They then subsequently describe Dlx genes as modulators of osteoclast activity and regulatory cascades involved during skeletal formation. This review eloquently describes the pivotal role Dlx genes play in chondrocyte and osteoblast differentiation and bone remodelling in adults, while describing how different levels of regulation of these genes may play a role in skeletal disorders and dis-regulated maintenance.

General concept comments [Review] Broad Comments/ Suggestions:

Overall the review is clear, comprehensive and relevant to the field. The statements drawn are coherent and supported well with publications. The sections are very well detailed, for example: section 2 describing the Distal-less gene family in vertebrates, it is very clear and well described.

The authors need to address a couple of points:

·      in the introduction sentence beginning The shape of bones and cartilages is genetically determined….. line 35”, I would argue that there is also mechanical influence to the determination of shape of the skeleton, as evidenced by a large body of work investigating this, addition of this reflected in this sentence would allow the authors to address this.

·      I think that figure 1 is very valuable, but the authors have not linked the descriptive text to it at all. I would recommend that they do so to utilise it fully.

Specific comments:

In section 4 Dlx2 role in chondrogenesis is described in detail solely from in vitro studies using a cell line, would it be of value to describe any embryonic expression that supports these observations. As with Dlx3 there is a large focus on in vivo expression and less on in vitro.

Note that a few grammatical issues are highlighted here to guide the authors

·      grammatical error on line 67

·      grammatical error on line 78, double plural?

·      grammatical error on line 199

·      reference required at the end of sentence 204.

·      reference required at the end of the sentence in line 232

·      grammatical error on line 280

·      reference needed at end of sentence on line 338

·      spelling error on line 350

Reviewer 3 Report

The manuscript provides an excellent summary of the role of Dlx genes in mammalian bone biology. The review is focused on the role Dlx genes in controlling bone formation and of bone cell (chondrocytes, osteoblasts, and osteoclasts) functions.

On the whole the manuscript is well-written though errors and specific suggestions are listed below.

Line 13: switch anabolism and catabolism

Line 33: change allowing to enabling

37: remove commas

40: delete “the”

41: change to “Pathological bone conditions can result from an imbalance in osteoblast-osteoclast communication.”

75: change “concerns” to “is the principal bone formation process for”

98: “delimitate”? How about “delineate”

100: change to “Runx2 also plays an…

110: change to osteoblast/osteoclast

113: change osteoclasts to osteoblasts

133: an homeodomain to a homeodomain

163: What does this mean? “, but only to a lesser extent in the mouse.”  This not clear. Do you mean that the penetrance of the imprinted allele is variable in mice?

180” change to .... might also act simultaneously in the adult…

201: change osteoblast to osteoblasts

324: The etiology of OA is unknown. That focal mechanical forces can induce articular cartilage degradation absent additional factors is unclear.  Please revise or clarify this statement.

326: change to: “Cartilage tissue is not vascularized and has a very limited regenerative potential. Thus cartilage injuries are associated with an increased risk for developing OA.” [There is no proof that the limited potential of cartilage to regenerate is because the tissue is avascular.]

338: change to: “Articular cartilage calcification indicates cartilage damage. The pharmacological inhibition of DLX5 by blocking antibodies….”

[This requires a little more information as it is unclear how an exogenous, extracellular antibody is inhibiting the activity of a transcription factor in the nucleus.]

342: “pa-pain-induced” What is this?

344: change to cartilage

349: change “at our” to “to our”

382: change to “As the rate of bone formation does not increase in vivo in these mice,…”

392: change to “increased trabecular bone mass in the Dlx3-deleted mice depends on enhanced…”

410: change to “…associated with familial cleft….”

417: change to: “Dlx5-/- mice die at birth. Histopathological analysis….”

419: change to “By E15.5…”

442: The sentence is unclear, particularly the clause “indeed, studies….” Please revise for clarity.

462: change to: “…of the DLX5/6 regulatory region identified 26 ….. capable of directing the expression of Dlx5/6 either…”

468: This sentence is unclear. Please revise for clarity.

477: change to: “region identified several”

479: change to “a haplotype”

489: ref 121 does not escribe any function for osteoclasts in fracture healing. Please find a more appropriate reference.

509: change to: As the osteoclast…..decreased in Runx2-/- mice, it seems…”

512: change “to” to “with”

513: change to: “…permit a better understanding of the origin…”

517: change to “…carrying conditional alleles of…”

589: spelling Chromsomal

626: change to “bone extract ChIPs”

649: Dx5 to Dlx5

677: This sentence is not clear. Please revise for clarity.

Round 2

Reviewer 1 Report

The changes made by the authors are persuasive